# Advances in Visible Light Communication Technologies and Applications

**Zuhang Geng** [1,2], **Faisal Nadeem Khan** [3,*], **Xun Guan** [3] **and Yuhan Dong** [1,2]

1    Tsinghua Shenzhen International Graduate School, Tsinghua University, Beijing 100084, China
2    Department of Electronic Engineering, Tsinghua University, Beijing 100084, China
3    Tsinghua-Berkeley Shenzhen Institute, Tsinghua Shenzhen International Graduate School,
     Tsinghua University, Beijing 100084, China
*    Correspondence: faisal.khan@sz.tsinghua.edu.cn

**Abstract:** With the development of light emitting diode (LED) lighting technology and its wide applications, visible light communication (VLC) technology has also seen significant advancements. VLC is regarded as a supplementary technology to radio frequency (RF) due to its unregulated spectrum and extraordinarily high communication rates. In this paper, the advantages, architectures, key technologies, application scenarios and machine learning (ML) applications of VLC are reviewed and summarized.

**Keywords:** visible light communication; machine learning; non-orthogonal multiple access; orthogonal frequency division multiplexing





## 1. Introduction

In recent years, the number of mobile devices has increased exponentially. However, the existing radio frequency (RF) might not fully satisfy the communication requirements in some scenarios due to its limited bandwidth, where intense competition for available radio resources leads to degradation of communication performance.

To solve the contradiction between limited bandwidth and the increased number of mobile devices, visible light communication (VLC) becomes a potential supplementary technology to RF. As shown in Figure 1, the wavelengh range of visible light is from 380 nm to 780 nm and thus, it has abundant bandwidth. As a result, VLC devices are capable of reaching much greater transmission speeds than RF devices. Furthermore, the spectrum of VLC is unregulated and it results in lower costs.

Another superiority of VLC over RF is the use of light emitting diode (LED) for data transmission. Residential LEDs have much higher power efficiency and longer lifespan than traditional incandescent light bulbs. VLC is regarded as a green communication technology with low energy consumption as it could satisfy the requirements of both illumination and communication simultaneously [1]. The wide employment of LED further reduces the cost of devices in VLC networks.

Due to the superiorities mentioned above, researches in VLC has exponentially increased in both theoretical techniques and practical applications. Several surveys in the past few years have been noted as well. For instance, H. Haas et al. review key advancements in physical layer techniques which significantly improve the transmission speeds of LEDs and discuss the challenges in achieving Tbps LiFi systems in [2]. L. E. M. Matheus et al. [3] discuss the main characteristics and future directions of VLC-based applications and the research platforms of VLC. An overview of optimization techniques to improve the performance of VLC networks is presented in [4] and it mainly focuses on how to apply the new technologies for RF networks to VLC networks. Additionally, the key performance metrics and recent achievements of hybrid LiFi and WiFi networks (HLWNets) are presented in [5],

which focuses on network architectures, cell deployments, multiple access and modulation schemes, illumination requirements and backhaul. A. Al-Kinani et al. [6] investigate the main optical channel characteristics and present a comprehensive overview of optical wireless communication (OWC) channel models. Furthermore, the channel models are compared in terms of computation complexity and accuracy. J. Luo et al. [7] discuss VLC-based indoor positioning and propose a novel taxonomy method. The VLC usage in vehicular communication applications is reviewed in [8,9], while the technology shortcomings and challenges are also presented. M. A. Arfaoui et al. [10] summarize the relevant technologies from the perspective of physical layer security (PLS) and point out the future research directions for PLS-VLC systems. Table 1 summarizes the surveys mentioned above and demonstrates the topics of the existing surveys and this paper.

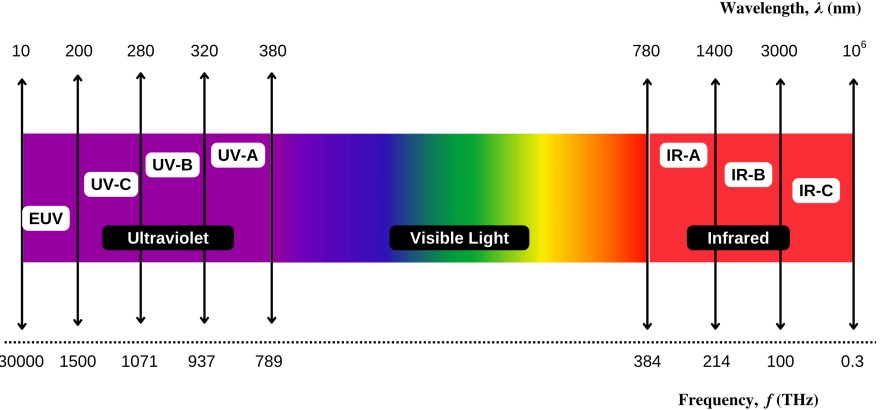

**Figure 1.** The optical spectrum.

**Table 1.** Surveys on Visible Light Communication.

| | Content Explored | H. Haas [2] | L. E. M. Matheus [3] | M. Obeed [4] | X. Wu [5] | A. Al-Kinani [6] | J. Luo [7] | A. -M. Căilean [8] | A. Memedi [9] | M. A. Arfaoui [10] | This Survey |
|---|---|---|---|---|---|---|---|---|---|---|---|
| | Superiorities of VLC | ✓ | ✓ | ✓ | ✓ | ✓ | ✓ | ✓ | ✓ | ✓ | ✓ |
| Architecture | Transmitter | ✓ | ✓ | ✓ | ✓ | ✓ | ✓ | ✓ | ✓ | ✓ | ✓ |
| | Receiver | ✓ | ✓ | ✓ | ✓ | ✓ | ✓ | ✓ | ✓ | ✓ | ✓ |
| | Transmission distance | | ✓ | ✓ | ✓ | ✓ | | ✓ | | | ✓ |
| | Channel modeling | ✓ | ✓ | ✓ | | ✓ | | ✓ | ✓ | ✓ | ✓ |
| Light modulation | OOK, PWM, PPM | | ✓ | ✓ | ✓ | | ✓ | | | | ✓ |
| | OFDM | | ✓ | ✓ | ✓ | | ✓ | | | | ✓ |
| Physical layer security | Keyless security techniques | | | ✓ | ✓ | | | | | ✓ | ✓ |
| | Key-based security techniques | | | ✓ | ✓ | | | | | ✓ | ✓ |
| | NOMA | | ✓ | ✓ | ✓ | | | | | ✓ | ✓ |
| | Machine learning | | | | | | | | | | ✓ |
| VLC applications | Indoor communication | ✓ | ✓ | ✓ | ✓ | ✓ | | | | | ✓ |
| | Positioning | | ✓ | | ✓ | | ✓ | ✓ | | | ✓ |
| | Vehicular communication | | ✓ | | | | | ✓ | ✓ | | ✓ |
| | Underwater communication | | ✓ | | | ✓ | | | | | ✓ |

However, most of the previous articles focus mainly on one of the research areas in VLC, which is not comprehensive enough. Additionally, to the best of our knowledge, none of the previous surveys deals with machine learning (ML). In this paper, we present a comprehensive survey on VLC as well as investigating the latest research progress of VLC and its applications in various emerging fields. The remainder of this paper is organized as follows. The architecture of VLC systems is presented in Section 2, while in Section 3, several key technologies of VLC are discussed, including channel modeling,

light modulation, PLS and non-orthogonal multiple access (NOMA). Additionally, ML algorithms applied in VLC systems are discussed as well. Section 4 provides the latest advances in VLC applications. Then, Section 5 concludes the paper.

## 2. Architecture of VLC Systems

Intensity modulation with direct detection (IM/DD) is usually adopted in VLC systems and LEDs are usually utilized as transmitters [11]. The whole architecture of a VLC system is shown in Figure 2. After the modulator and pulse shaper, the transmitted information is loaded on an electrical signal, which is going to be converted into an optical signal at the transmitter. At the receiver side, the photodetector generates an electrical signal according to the intensity of the received optical signal. The inherent interference caused by the ambient light and the multi-path problem will lead to the reduction of transmission performance. As a result, bandpass filters and amplifiers are adopted to restore the time-domain signals. Then, the original information is recovered by the demodulator.

The visible light signals might be scattered by small particles in atmospheric or underwater channels due to its relatively low wavelengths. To solve this problem, a lens is adopted to collect and focus the received beam onto the photodiode (PD) at the receiver side [12,13]. Furthermore, multi-hop VLC is proposed to extend the communication range and it is regarded as a solution to the main issues including attenuation, scattering and divergence [14–16].

The transmission distance and data rate are limited by the channel conditions, light sources and photodetectors. In atmospheric channels, the transmission distance varies from few meters to few kilometers [16–20]. In contrast, the transmission distance is usually restricted to 500 meters in underwater channels as a result of the significant attenuation [21–23].

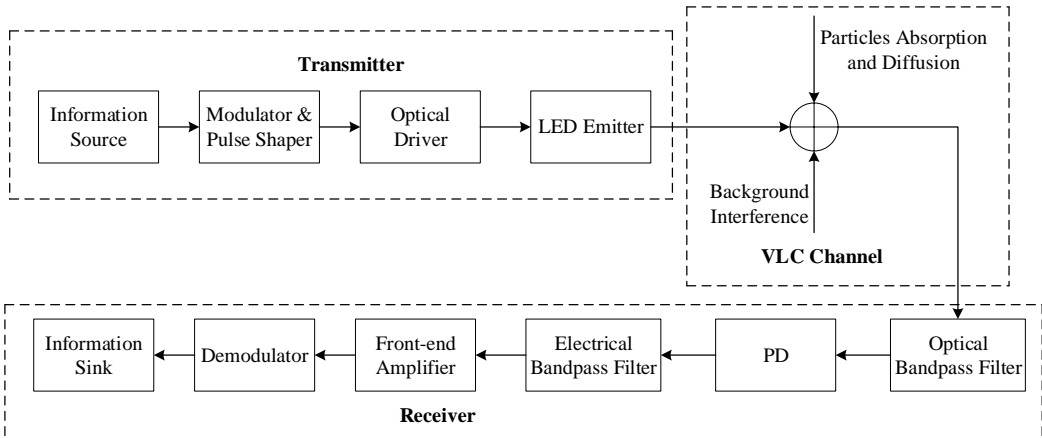

**Figure 2.** Architecture of a VLC system.

## 3. Key Technologies for VLC Systems

Considering the fact that the optical channel suffers from various noise and interference effects, techniques in physical and media access control (MAC) layers are proposed to ensure the desired communication quality. For example, research on channel characterization could help to reduce the influences caused by the noise and interference. Light modulation helps to utilize light intensity to convey data bits. Furthermore, research in physical layer security ensures the communication security of VLC systems. In the MAC layer, multiple access (MA) is adopted in many VLC scenarios to support multiple devices connected simultaneously and NOMA significantly improves the resource efficiency compared to traditional schemes. Furthermore, ML is proposed as a supplement of traditional algorithms in VLC systems.

### 3.1. Channel Characterization

In some VLC scenarios, the complex link and the dynamically changing channel state lead to challenges in signal transmission and system design. To solve this problem, research in VLC channels is required to obtain real-time channel status. In VLC systems, geometrical optics is usually adopted for channel modeling due to the short wavelength of visible light [8] and the channel variation is mainly caused by the translation and rotation of the receiver, which is different from conventional RF channels. The indoor VLC propagation scenario including both line of sight (LOS) and non-LOS (NLOS) paths is shown in Figure 3.

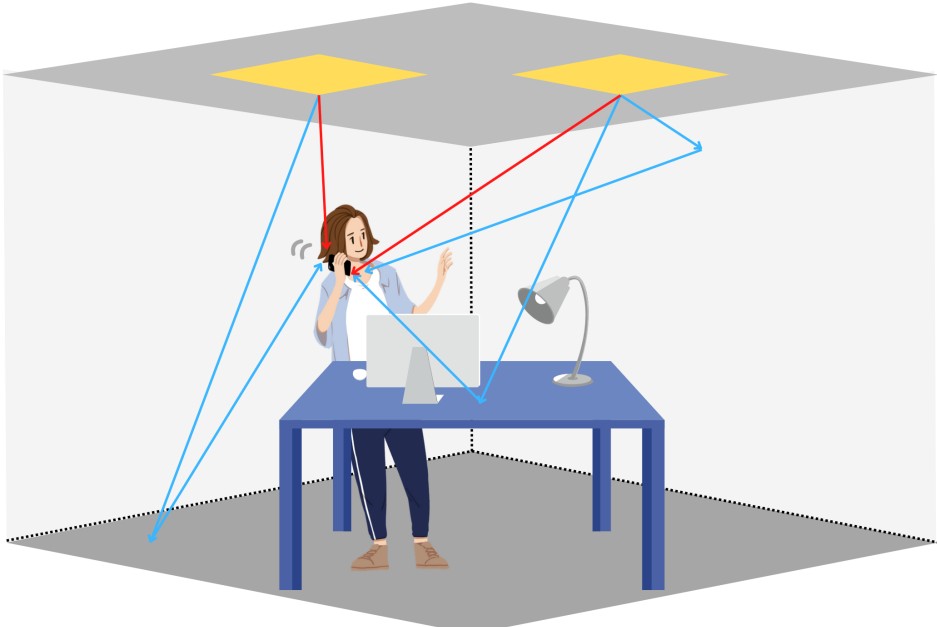

**Figure 3.** Indoor VLC propagation scenario.

As a key feature of VLC channel, the instantaneous impulse has been researched in [24,25]. However, VLC channels become time-varying with the movement or rotation of the receiver. As a result, methods for dynamic channel modeling are required to obtain the instantaneous channel state information (CSI), which is proved to be crucial to improving system throughput in [26,27].

J. Chen et al. proposed a movement–rotation (MR) correlation function to measure VLC channel variations for LOS channel in [28]. The MR correlation function could be utilized to measure VLC channel variations without time dependence as the receiver moves and rotates. Simulation results show that receiver rotations usually result in small channel fluctuations, while receiver movements lead to large-scale channel variations. The authors utilize the MR correlation function to approximate the correlation function of VLC channel gain and analyze the system performances for varying VLC channels, which is significant for the design and analysis of adaptive data transmission.

Another channel model for VLC is presented in [29], where X. Zhu et al. propose a novel three-dimensional (3D) space-time non-stationary geometry-based stochastic model (GBSM) for indoor multiple input multiple output (MIMO) VLC channels. This is one of the first VLC GBSM to support 3D translational and rotational motions, special radiation patterns of LEDs and space-time-frequency non-stationarity. The authors investigate several key statistical properties including channel DC gain, received power, channel 3dB bandwidth, space-time-frequency correlation function (STFCF) and root mean square (RMS) delay spread. The proposed GBSM is a better fit to the measured data than existing models, which confirms the accuracy and practicality of it.

It could be seen that the latest studies mainly focus on the dynamic changes of the channel state based on existing mathematical functions or models. It plays an important

role in analyzing the dynamic characteristics of the channel, reducing the influence caused by noise and interference and designing new VLC systems.

*3.2. Light Modulation*

As mentioned above, one of the most significant differences between VLC and RF is that the transmitted data have to be encoded in the optical signals with different intensities. There are two main limitations for light modulation in VLC. Firstly, dimmer circuits are equipped on light bulbs to provide the ability to control light intensity. Secondly, no human-perceivable fluctuations are allowed in the modulated light waves in order to prevent serious detrimental physiological damage to humans [30].

The traditional light modulation schemes include on-off keying (OOK), pulse width modulation (PWM), pulse position modulation (PPM) and orthogonal frequency division multiplexing (OFDM). OOK mudulation turns the LED on and off to transmit the data bits 1 and 0, respectively. It is easy to implement yet it suffers from flickering and limited data rate. PPM loads the transmitted data on the position of the pulse while the length of the pulse corresponds to the value of the signal in PWM. As a result, PWM and PPM are capable of transmitting data without the variation of the pulse intensity.

Recently, several PWM schemes have been proposed for VLC systems. In order to increase the bandwidth usages for VLC, M. A. S. Sejan et al. [31] propose multilevel PWM (MPWM). The suggested modulation technique can impose additional bits by adopting pulse height and width modifications simultaneously. However, the increase of pulse height and width levels leads to higher BER performance than traditional schemes.

In previous studies, rectangular waves are usually assumed as the underlying received signal waveforms. However, uninterrupted on/off switching of electronic elements leads to non-rectangular waves, which is demonstrated in [32]. K. Yan et al. [33] propose a new precise PPM-VLC received-signal model considering the non-rectangular waves. Based on the PPM-VLC received-signal model, a new PPM-VLC demodulation scheme is proposed. Furthermore, a package-template of an L-PPM symbol is constructed and the new modulator evaluates the similarity between the package-template and the received signal waveform to recover the transmitted information. Therefore, it solves the problem of dynamic threshold adjustment, which implies better performance than a traditional demodulator.

Additionally, the four-level pulse amplitude modulation (PAM4) format has been extensively researched and experimentally proven to be an appealing technique to boost spectral efficiency. However, optical multipath interference (MPI) noises extremely decrease the transmission quality of IM/DD systems [34–37]. C. Huang et al. propose two algorithms for MPI noise elimination in [38] by removing the fluctuation of MPI-impaired PAM4 signals and estimating the MPI noise, respectively. Without altering the current IM/DD system design, these algorithms could be implemented in the receiver digital signal processing (DSP) module directly. Furthermore, the simulation and experimental results demonstrate their capacity for suppressing the MPI noise for IM/DD transmission systems with high-speed and high-order modulation formats.

Given that single-carrier modulation schemes suffer from high inter-symbol interference (ISI), OFDM is adopted in VLC with the advantage of high spectral efficiency and characteristics for resisting ISI and multipath fading [39]. However, the traditional OFDM time-domain signal in RF is complex-valued and bipolar, which is contradictory to IM/DD systems [11]. Generally, Hermitian symmetry is adopted to generate real-valued time-domain signals. In order to obtain unipolar signals, several optical OFDM (O-OFDM) schemes have been proposed, which are summarized in Table 2. The most popular O-OFDM schemes are asymmetrically clipped optical OFDM (ACO-OFDM) and direct-current (DC) biased optical OFDM (DCO-OFDM). ACO-OFDM only occupies odd subcarriers and clips the negative part of time-domain signals with no information lost [40]. Whereas, in DCO-OFDM, a DC bias is added to make the time-domain signal nonnegative [41]. Another O-OFDM scheme named pulse-amplitude-modulated discrete multitone (PAM-DMT) only utilizes the imaginary part of subcarriers and clips the negative part of time-domain signals

as well as ACO-OFDM [42]. Based on the O-OFDM schemes mentioned above, several variants are proposed in succession. For instance, hybrid ACO-OFDM (HACO-OFDM) combines both ACO-OFDM and PAM-DMT [43], while layered ACO-OFDM (LACO-OFDM) transmits layers of ACO-OFDM simultaneously and recovers the transmitted data iteratively [44]. LACO-OFDM has much higher spectral efficiency and lower peak-to-average power ratio (PAPR) than ACO-OFDM and HACO-OFDM.

**Table 2.** Summary of Optical OFDM Schemes in VLC.

| Work | O-OFDM Scheme | Utilized Spectral Resource | Signal Processing | Features |
|---|---|---|---|---|
| J. Armstrong et al. [40] | ACO-OFDM | Odd subcarriers | Clipping operation | Low complexity, low spectral efficiency and high power efficiency |
| J. B. Carruthers et al. [41] | DCO-OFDM | All subcarriers | Adding a DC bias | Low complexity, high spectral efficiency and low power efficiency |
| S. C. J. Lee, et al. [42] | PAM-DMT | The imaginary part of subcarriers | Clipping operation | Low complexity, low spectral efficiency and high power efficiency |
| B. Ranjha et al. [43] | HACO-OFDM | Odd subcarriers and the imaginary part of even subcarriers | Clipping operation | Higher spectral efficiency than ACO-OFDM |
| Q. Wang et al. [44] | LACO-OFDM | Layers of the half of the remained subcarriers | Clipping operation | Higher spectral efficiency than HACO-OFDM and an iterative receiver with higher complexity |
| R. Bai et al. [45] | AAO-OFDM | All subcarriers | Clipping operation and absolute operation | Higher spectral efficiency than ACO-OFDM |
| R. Bai et al. [46] | ALACO-OFDM | All subcarriers | Clipping operation and absolute operation | Higher spectral efficiency than AAO-OFDM and an iterative receiver with higher complexity |

Asymmetrically clipped absolute value optical OFDM (AAO-OFDM) is proposed in [45], it utilizes two streams to send ACO-OFDM signal and absolute value optical OFDM (AVO-OFDM) signal, respectively. The signs of the AVO-OFDM signals are modulated to the frequency-domain symbols of ACO-OFDM. As a result, signals of AVO-OFDM do not require any DC bias to generate unipolar values. Furthermore, in [46], absolute value layered asymmetrically clipped optical OFDM (ALACO-OFDM) is proposed, and it reaches higher spectral efficiency by transmitting ACO-OFDM signals in the first $L$ layers and absolute value optical OFDM (AVO-OFDM) signals on the remaining subcarriers simultaneously. Taking uncoded bit-error-ratio (BER) and achievable information rate into account, R. Bai et al. [46] designed two optimal optical power allocation schemes, respectively. Additionally, analysis shows that ALACO-OFDM achieves a higher information rate at moderate to high signal-to-noise ratios (SNRs) and has lower PAPR than ACO-, AAO- and LACO-OFDM.

Furthermore, signals are modulated using the indices of a medium in the index modulation (IM) approach to further enhance the spectral efficiency or power efficiency of O-OFDM systems [47]. By utilizing discrete Hartley transform (DHT), X. -Y. Xu et al. [48] propose a new O-OFDM-IM scheme with lower complexity and higher spectral efficiency than traditional O-OFDM schemes based on discrete Fourier transform (DFT). It is superior in SNR performance as well. However, the PAPR performance of this scheme is not satisfying compared with its traditional counterparts sometimes.

In this section, we discussed several modulation schemes, which generate unipolar real-valued signals and meet the requirements of IM/DD systems. OOK, PWM and PPM are simple to realize while OFDM could reduce ISI and is suitable for MIMO. The best scheme should be chosen according to the specific scenario.

### 3.3. Physical Layer Security

Considering the fact that light does not penetrate through walls, VLC has higher security than RF systems. However, unlike fiber-optic systems, security problems might occur in VLC systems owing to their open and broadcast nature [49]. PLS techniques have been fully studied and applied in RF systems [4,50]. The two main categories of PLS techniques are keyless security techniques and key-based security techniques. As a result of the differences between RF and VLC, PLS technologies developed for RF systems may not be directly applicable to VLC systems.

Keyless security techniques usually utilize the randomness of noise, channels and different resources to enhance the security of networks [50]. N. Su et al. [51] propose a novel spatial constellation design technique based on a multi-user generalized space shift keying for indoor multi-user MIMO-VLC (MU-MIMO-VLC) scenario to enhance the PLS. The authors adjust the transmission power of each transmitting LED by adjusting the CSI of legitimate users, which optimizes the received signal constellation for legitimate users in terms of BER and only generates interference for the eavesdroppers. Simulation results demonstrate that the BER of an eavesdropper declines significantly and the confidentiality improvement depends on the relative position between users.

Y. M. Al-Moliki et al. [52] propose a chaos-based physical-layer encryption method for OFDM-based VLC schemes. By using the position-sensitive and real-valued CSI of the VLC channel, a chaotic key creation approach is developed to create the secret key. However, based on floating-point arithmetic, chaotic systems have a significant resource and latency overhead, which is inappropriate for resource-limited VLC devices and high-data-rate VLC systems. Furthermore, in Y. M. Al-Moliki et al. [53] propose a key-based lightweight channel-independent (LCI) physical-layer encryption method, which generates dynamic keys and ciphertexts with the random nature of the input data and applies phase encryption of OFDM symbols in the frequency domain. The proposed method is suitable for resource-restricted scenarios and has relatively low complexity. Another lightweight cipher scheme for VLC systems is proposed in [54]. The proposed approach secures the underlying OFDM signals using straightforward substitution and phase shuffling techniques with low computational complexity and latency.

The key-based security techniques mentioned above are superior to keyless security techniques in design complexity. However, keyless security techniques could provide better security than key-based security techniques. Through advanced coding techniques at the physical layer, PLS has the potential to take advantage of elements of the environment around it [55], which can contribute to realizing the robust end-to-end security and satisfying the requirements of the next generation networks.

### 3.4. NOMA

In VLC systems, traditional MA techniques, including frequency division multiple access (FDMA), time division multiple access (TDMA) and code division multiple access (CDMA), suffer from low resource efficiency [56]. To solve this problem, orthogonal frequency division multiple access (OFDMA) is proposed to reuse subcarrier resources. H. Marshoud et al. [57] propose NOMA in order to further enhance resource efficiency and system capacity in VLC systems, aiming at increasing the throughput, reducing the latency and improving the fairness and connectivity. Multiple users' signals are superimposed in the power domain and each user could utilize the entire time and frequency resources. In NOMA, users with poor channel conditions are assigned more signal power, while users with good channel conditions corresponds to less power. The transmitted information is recovered by successive interference cancellation (SIC) at the receiver side.

Considering uniformly distributed users, L. Yin et al. [56] derive the distribution function of the channel gain in a closed form. Additionally, the performance of NOMA is evaluated and compared with orthogonal multiple access (OMA) and a closed-form expression of the ergodic sum rate gain of NOMA over OMA is derived. In [58], the BER performance in a downlink NOMA-VLC network is analyzed and an exact, simple and generic analytic expression is derived for the BER performance, which is the first work to study the BER performance of NOMA-based systems. The performance of a hybrid NOMA-VLC-RF system with imperfect CSI and uniformly distributed users is evaluated in [59]. The authors derive closed-form expressions for the corresponding average sum-rate and average energy efficiency, which coincide with the simulation results. M. Le-Tran et al. [60] analyze the NOMA performance in a downlink VLC system with an optical backhauled link and derive the closed-form expressions of the user outage probability, the sum throughput, the average BER and the energy efficiency with guaranteed transmission

rates. The theoretical and simulation results imply that NOMA systems provide significant performance gains for high-rate optical backhaul links compared with OMA systems at medium to high SNR ranges. The first non-OFDM-based NOMA scheme for VLC with arbitrary modulation order in multiple access and broadcast channel is proposed in [61], which is appropriate for high-SNR regimes and requires lower computational complexity than the existing schemes.

In NOMA, all time and frequency resources are shared by all users with different power. It offers a higher quality of service and better capacity for resisting interference, and it is regarded as a promising technique.

### 3.5. Machine Learning

In the past few years, ML has attracted intense interest of researchers and has been regarded as a potential technology to solve the various challenges in wireless communication systems. ML performs well in resources allocation, channel equalization, estimation and modeling [62]. As a result, the applications in VLC are exponentially increasingly proposed. For instance, M. Najla et al. [63] study the selection between RF and VLC bands for device-to-device (D2D) communication, assuming the condition that sudden drops in channel quality occur in VLC links. A deep neural network (DNN)-based framework to select RF or VLC for D2D pairs is proposed to obtain an initial band selection decision. The authors further present a low-complexity heuristic algorithm to improve the accuracy of the band selection and simulation results show the close-to-optimal performance of the proposed algorithm.

To overcome the deterioration in communication and positioning performance resulting from the long transmission distance in vehicular VLC (V-VLC), J. He et al. [64] propose and experimentally demonstrate an ML-assisted image sensor-based visible-light-based positioning (VLP) scheme. At the transmitter side, a new coding method is adopted to increase the data rate and adjust the LED lighting power according to the ambient light intensity. At the receiver side, a convolutional neural network (CNN) and an artificial neural network (ANN) are used for decoding and vehicle positioning, respectively, which help to realize long-distance communication, high-accuracy positioning and LED dimming simultaneously.

The issue of dynamically deploying unmanned aerial vehicles (UAVs) in VLC for improving the energy efficiency of UAV-enabled networks is investigated in [65]. In order to jointly maximize the usage of UAVs, user association and power efficiency while satisfying user lighting and communication needs, an approach is developed to address this issue by combining the ML framework of gated recurrent units (GRUs) with CNNs. The proposed method can forecast future light distributions by modeling long-term historical illumination distributions, which significantly reduces power consumption when compared to traditional methods, according to simulation results.

Collaborative constellation (CC) design is useful for enhancing performance while drastically lowering the total optical power in MIMO VLC systems. M. Le-Tran et al. [66] propose CCNet, a DL-based constellation design technique for MIMO VLC systems with CC that can drastically reduce complexity while preserving near-optimal performance when compared to previous schemes. CCNet is initially trained offline to decrease the mean square error (MSE) and the ordinary CSI is effectively preprocessed to further enhance the performance of CCNet.

In order to enable efficient resource management in VLC, Z.-Y. Wu et al. [67] proposed a data-driven ML-based approach to forecast LOS link outages and minimize severe signal degradation. Furthermore, a predictor is designed to learn the channel variation patterns and predict LOS link outages and recoveries by utilizing a deep recurrent neural network (RNN) with long-short-term-memory (LSTM) units. For both uplink and downlink, the proposed predictor achieves a 91% hit rate for outages and an 83% hit rate for signal recoveries when predicting the channel in the next one second. As a result, the development

of effective resource management strategies in VLC networks could be greatly aided by this predictor.

Table 3 summarizes part of the latest research advances of ML in VLC systems. Research on ML has been growing exponentially recently and it will provide significant support for the advancement of VLC technology.

**Table 3.** Summary of ML applications in VLC.

| Work | Application Scenarios | Neural Network | Advantages |
| --- | --- | --- | --- |
| Mehyar Najla et al. [63] | The selection between RF and VLC bands for D2D communication | DNN | Close-to-optimal performance |
| Jing He et al. [64] | Long-distance transmission in V-VLC | CNN and ANN | Long-distance communication, high-accuracy positioning and LED dimming |
| Yining Wang et al. [65] | Dynamically deploying UAVs in VLC | CNN | Future light distribution forecast and low power consumption |
| Manh Le-Tran et al. [66] | Collaborative constellation design | DNN | Low complexity and near-optimal performance |
| Zi-Yang Wu et al. [67] | Prediction of LOS link outage | deep LSTM-based RNNs | High hit rate for signal outages and recoveries |

## 4. VLC Applications

VLC has a wide range of applications in various fields. Thanks to the characteristics of VLC, it could achieve many functions that cannot be realized by RF. For instance, VLC performs much better than RF in scenarios that require a high data rate, such as underwater high-speed video communication. Additionally, VLC is capable of meeting the demand for illumination and communication simultaneously, while RF is not applicable evidently.

In this section, we will discuss the research advances in the past few years that mainly focus on indoor communication, positioning, vehicular communication applications and underwater communication.

### 4.1. Indoor Communication

With the wide application of LED bulbs, more and more attention has been paid to the research of indoor VLC systems. M. A. Arfaoui et al. [68] propose realistic and measurement-based channel models for indoor VLC systems. The modified truncated Laplace (MTL) model and the modified Beta (MB) model are designed for stationary users, while the sum of modified truncated Gaussian (SMTG) model and the sum of modified Beta (SMB) model are proposed for mobile users.

Another significant issue is the orientation variation of terminals in VLC. A. A. Purwita et al. [69] propose a random process model for user equipment (UE) orientation variation and present how it affects the optical channel conditions. The results demonstrate that the blockage and diffuse connection have considerable consequences, particularly when the UE is situated far from an access point (AP).

In terms of system implementation, C. -H. Yeh et al. [70] achieve a data rate of 1.7 to 2.3 Gbps with a communication distance of 1 to 4 m. The illumination is extremely low and is set to be 6.9 to 136.1 lux. Two blue and two green LEDs are utilized and a $4 \times 4$ color-polarization-multiplexing method is proposed. Additionally, to maintain the improved signal performance, the measured BER of each LED is lower than the forward error correction objective of $3.8 \times 10^{-3}$.

### 4.2. Positioning

The Global Positioning System (GPS) has been widely used to provide real-time positioning and navigation. However, signals transmitted by satellites are usually weakened by obstacles, which reduces the accuracy of indoor positioning [71,72]. WiFi and Bluetooth positioning systems are usually adopted to improve the performance of indoor positioning in the previous works [73,74]. In recent years, positioning systems utilizing a visible light signal rather than RF are proposed. A typical prototype of an LED-based indoor positioning

system is shown in Figure 4 where several LEDs are utilized as transmitters and provide lighting. Received signal strength (RSS), time-of-arrival (TOA), time difference of arrival (TDOA) or angle of arrival (AOA) information at the receiver side could be utilized to evaluate the localization. Moreover, the position coordinates could be obtained directly from captured images if the PDs are replaced by cameras [75]. The general architecture of a simplex VLC positioning system is presented in Figure 5.

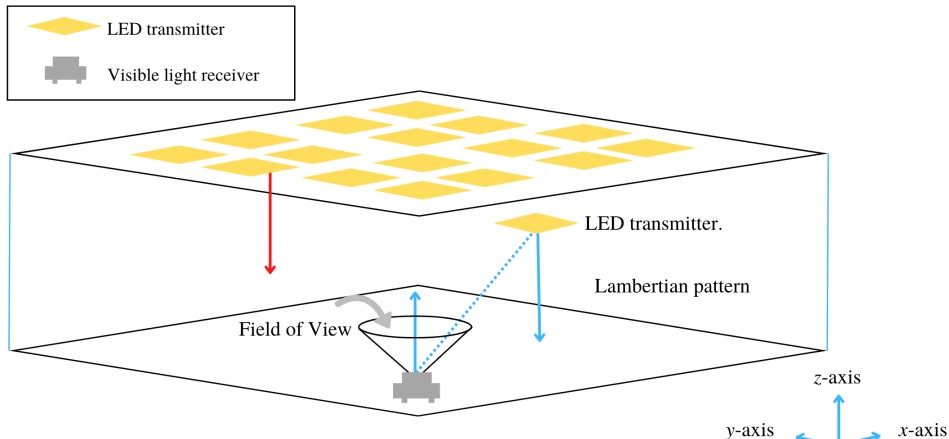

**Figure 4.** Illustration of the VLC-based positioning system.

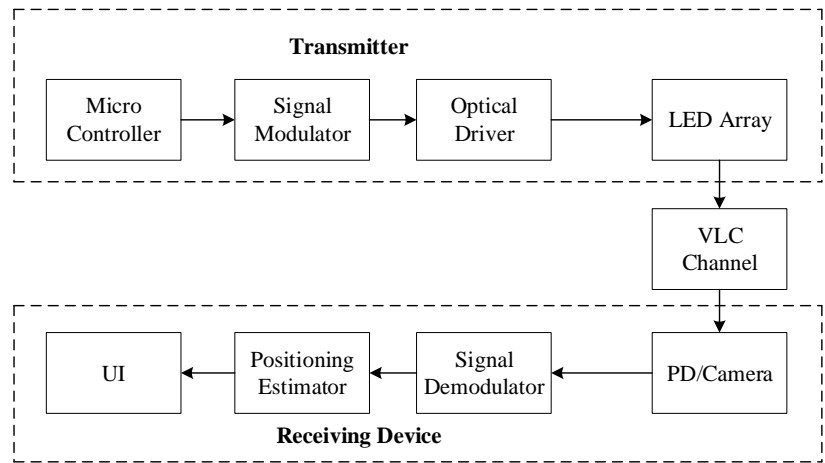

**Figure 5.** The general architecture of a simplex VLC positioning system.

B. Zhou et al. [76] propose a new VLC localization algorithm with the assumption of unknown LED emitting power, UE position and UE orientation. The joint optimization of all unknown parameters is adopted and a successive linear least square (SLLS)-based VLP algorithm is proposed. The authors derive the closed-form Cramer–Rao lower bound (CRLB) on each unknown parameter and analyze the performance limits of the proposed algorithm. Based on improved hybrid bat algorithm (IHBA), Y. Chen et al. [77] propose an indoor VLC 3D positioning system. The simulation results demonstrate higher positioning accuracy and shorter convergence time of IHBA compared with the existing VLC 3D positioning algorithm.

A position estimation DNN (PE-DNN)-aided receiver is proposed in [78], which utilizes the received pilot signals to extract the feature of the channel impulse response (CIR). Then, the coordinates are obtained from the CIR and an LED and a PD is sufficient to achieve centimeter-level positioning accuracy. Furthermore, it could achieve information transmission and positioning simultaneously, which ensures the compatibility and practicality of this VLC system.

Indoor positioning with VLC is regarded as a potential complement to GPS and it is capable of providing more accurate localization. Nevertheless, obstacles and reflection components might reduce the accuracy of positioning and these issues remain to be studied.

### 4.3. Vehicular Communication

Transportation systems have recently been developing by leaps and bounds. Intelligent transportation systems (ITS) are discussed in [79] and information exchange between vehicles and with infrastructure is indispensable for achieving ITS, where RF communication plays an important role. However, with the wide range of LED adoption, V-VLC could be realized by utilizing the LED-equipped lighting modules and transportation infrastructure [8].

Figure 6 presents a typical V-VLC traffic scenario, which includes infrastructure to vehicle communication, head-to-tail, tail-to-head, head-to-head and tail-to-tail communication. The building blocks of a generic V-VLC system are shown in Figure 7. Encoder, modulator, LED driver and optical transmitter front-end are equipped at the transmitter side, while optical receiver front-end, demodulator and decoder are equipped at the receiver side, correspondingly. The optical channel is interfered by other light sources, weather conditions and reflections.

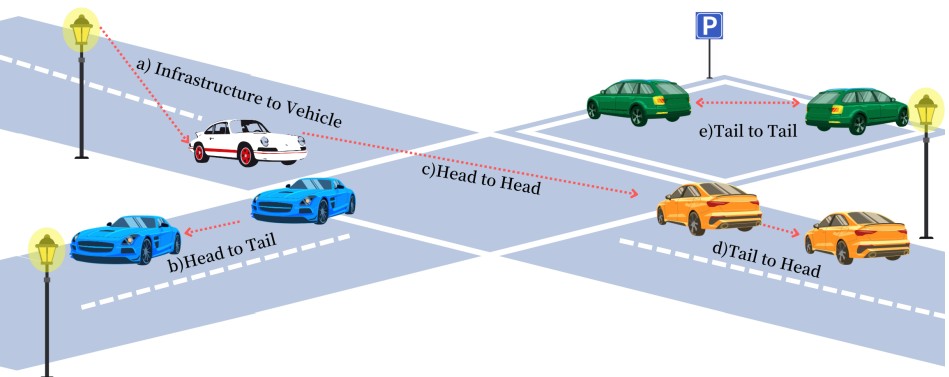

**Figure 6.** A typical V-VLC traffic scenario.

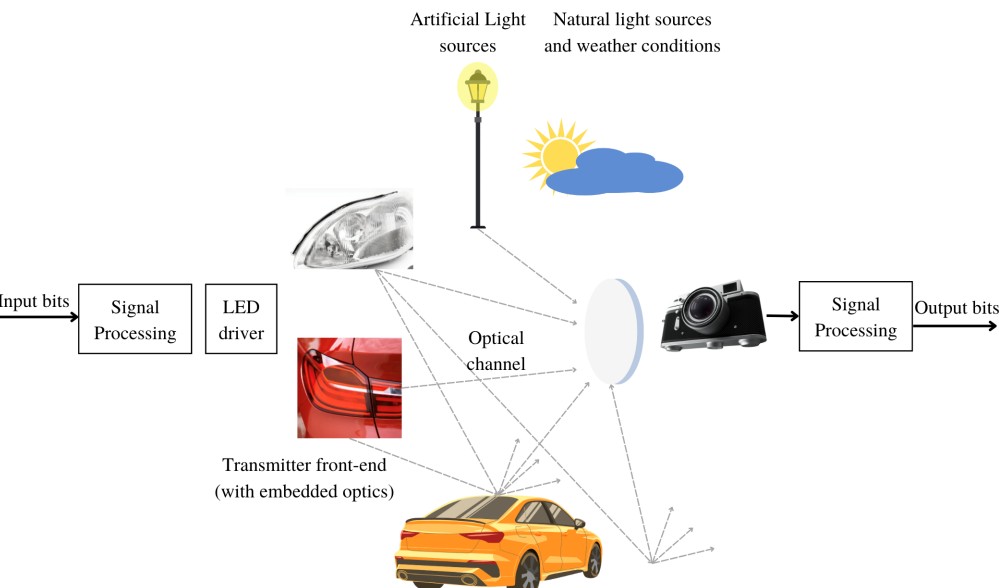

**Figure 7.** Building blocks of a generic V-VLC system.

Platooning is a key scenario for autonomous driving, where vehicles utilize vehicle-to-vehicle (V2V) communication and distance sensors to automatically adjust their position. It

has gained strong interest in realizing illumination, data transmission and range-finding simultaneously with the automotive lighting. In [80], a system named visible light communication rangefinder (VLCR) is proposed. The V2V distance is estimated by utilizing the phase-shift between the original signal and the received signal, where the Doppler effect is found to be neglected. Experimental results show that the range-finding function could work at up to 25 m and the system is able to support a 500 kbps link with a BER below $10^{-6}$ and a transmission distance of up to 30 m.

Based on GNU Radio, M. S. Amjad et al. [81] introduce a flexible IEEE 802.11 compliant system for outdoor V-VLC and a high-power LED headlight is adopted to support communication distances beyond 75 m even in broad daylight. The authors also study the impact of optics alignment on the receiver's performance and analyze how the daylight influences the PD noise floor.

In [82], a vehicular MIMO VLC system based on two commercial headlights and a self-designed PIN array is presented and the strategy for selecting the best MIMO de-multiplexing scheme by analyzing the rank and type of the channel matrix is discussed. Additionally, the authors proposed a modified pilot-aided phase recovery method based on polynomial curve fitting (PCF) and a record-breaking data rate of 3.08 Gbps at a 2 m indoor transmission link is realized. Furthermore, the overall data rates reach 336 Mbps and 362 Mbps in the day and at night, respectively, when the transmission distance is extended to 100 m, which are the highest transmission data rates of a vehicular MIMO VLC system for a 100 m transmission distance until now.

A major disadvantage of VLC is that the headlights and taillights are not capable of communicating directly with side-to-side vehicles, which demonstrates the lack of preventing blind-spot oversight [83]. Moreover, stringent latency and reliability are required for the security of vehicle driving in V-VLC and the performance of V-VLC is influenced by the outdoor environment, which bring challenges for V-VLC applications.

### 4.4. Underwater Communication

Underwater wireless communication (UWC) refers to data transmission via wireless carriers, i.e., RF waves, acoustic waves, and optical waves in unguided water environments. The characteristics of the underwater channel present many challenges compared to traditional wired and wireless communications through the atmosphere. Higher transmission bandwidth and data rates make underwater optical wireless communication (UOWC) more suitable than RF and acoustic counterparts for UWC systems [12]. Additionally, the turbulence of underwater environments results in changes in water density and salinity, which may reduce the performance of UOWC systems. It has been proved in [84] that flicker index and BER are significantly influenced by average temperature, average salinity concentration, temperature–salinity gradient ratio, temperature dissipation rate and energy dissipation rate.

Many researchers focus on improving transmission data rate and extending communication distance in UOWC experimental systems. For instance, X. Yang et al. [21] utilize the arrays consisting of series-connected monochromatic LEDs to reach a data rate of 130 Mbps over a 7 m underwater channel in LED-to-LED UOWC systems. Moreover, X. Chen et al. [22] realized a data rate of 500 Mbps with a transmission distance up to 150 m by combining partial response shaping and trellis-coded modulation (TCM) technology for the first time. Furthermore, H. Zhou et al. [23] propose a new mathematical model of UOWC. It could be used for water quality measurement and the proposed system could reach a 50 m link with the data rate of 80 Mbps.

With the exponentially increasing research in VLC, it will be utilized as a complement to RF in more and more scenarios to improve the performance of communication systems. However, challenges including flickering, dimming, noise and interference remain a significant impediment to the widespread adoption of VLC, which requires further research.

## 5. Conclusions

In this work, we thoroughly investigate the literature on VLC in recent years. With a high data rate and unlicensed spectrum, VLC is regarded as an excellent alternative to RF to satisfy the increasing demand for wireless resources. Related techniques are researched in physical and MAC layers to reduce the impact of interference and ensure desired communication performance. Additionally, ML has been adopted in VLC systems and the related research has increased exponentially, which shows a new direction for VLC research.

VLC has a wide range of applications in many short-range communication scenarios, such as indoor communication, indoor positioning, vehicular communication and underwater communication. However, there are still many issues and challenges in the application of VLC technology in multiple scenarios, such as flickering, long-distance transmission and interference. Many promising VLC technologies are not yet well developed, and this research area needs further investigation.

**Author Contributions:** Conceptualization, Z.G., F.N.K., X.G. and Y.D.; writing—original draft preparation, Z.G.; writing—review and editing, F.N.K., X.G. and Y.D.; supervision, F.N.K., X.G. and Y.D.; project administration, F.N.K.; funding acquisition, F.N.K., X.G. and Y.D. All authors have read and agreed to the published version of the manuscript.

**Funding:** The work was supported in part by the GuangDong Basic and Applied Basic Research Foundation under Grant 2022A1515010209 and Shenzhen Natural Science Foundation under Grant JCYJ20200109143016563. Faisal Nadeem Khan would like to acknowledge the support of Tsinghua Shenzhen International Graduate School and Tsinghua–Berkeley Shenzhen Institute under Scientific Research Startup Fund (Project number: 01010600001(CD2022004C)).

**Institutional Review Board Statement:** Not applicable.

**Informed Consent Statement:** Not applicable.

**Data Availability Statement:** Not applicable.

**Conflicts of Interest:** The authors declare no conflict of interest.

## Abbreviations

The following abbreviations are used in this manuscript:

| | |
|---|---|
| VLC | Visible light communication |
| RF | Radio frequency |
| ML | Machine learning |
| LED | Light emitting diode |
| OWC | Optical wireless communication |
| MAC | Media access control |
| PLS | Physical layer security |
| MA | multiple access |
| NOMA | Non-orthogonal multiple access |
| IM/DD | Intensity modulation with direct detection |
| LOS | Line of sight |
| NLOS | Non-line of sight |
| CSI | Channel state information |
| MR | Movement-rotation |
| GBSM | Geometry-based stochastic model |
| MIMO | Multiple input multiple output |
| STFCF | Space-time-frequency correlation function |
| RMS | Root mean square |
| OOK | On-off keying |

| | |
|---|---|
| PWM | Pulse width modulation |
| PPM | Pulse position modulation |
| OFDM | Orthogonal frequency division multiplexing |
| PAM4 | 4-level pulse amplitude modulation |
| MPI | Multipath interference |
| DSP | Digital signal processing |
| SNR | Signal-to-noise ratio |
| IM | Index modulation |
| DHT | Discrete Hartley transform |
| DFT | Discrete Fourier transform |
| LCI | Lightweight channel-independent |
| SIC | Successive interference cancellation |
| OMA | Orthogonal multiple access |
| DNN | Deep neural network |
| CNN | Convolutional neural network |
| ANN | Artificial neural network |
| UAV | Unmanned aerial vehicle |
| GRU | Gated recurrent unit |
| CC | Collaborative constellation |
| MSE | Mean square error |
| RNN | Recurrent neural network |
| LSTM | Long-short-term-memory |
| GPS | Global positioning system |
| RSS | Received signal strength |
| TOA | Time-of-arrival |
| TDOA | Time difference of arrival |
| AOA | Angle of arrival |
| PD | Photodiode |
| UE | User equipment |
| VLP | Visible light-based positioning |
| CIR | Channel impulse response |
| ITS | Intelligent transportation systems |
| V-VLC | Vehicular VLC |
| D2D | Device-to-device |
| V2V | Vehicle-to-vehicle |
| PCF | Polynomial curve fitting |

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
