# Peer review of "Advances in Visible Light Communication Technologies and Applications"

_photonics, doi:10.3390/photonics9120893_

Round 1

Reviewer 1 Report

This paper surveyed the advantages, architectures, technologies and machine learning (ML) applications in VLC. Although, the study has limited contribution, the work can be recommended for publication after addressing the following major critical concerns:  

·        Paper contributions and novelty compared to the existing surveys are not clear. A literature review matrix and comparison are required to highlight areas of contributions against similar works.

·        The paper discussed basic VLC concepts without addressing details about state-of-the-art technologies in VLC. It also surveyed limited references without considering high-impact studies in this field. Extensive literature review is required to cover recent areas in VLC.

·        A critical review is required. In many paragraphs, the discussion summarized the methodologies of previous works without indicating the outcomes and gaps in these studies. (e.g., third paragraph section I page 1, and section 3.4).

·        References are required to be cited in the manuscript in order of appearance. For example, in section 2 reference [28] appears after [10] directly. Similarly, [59-64] appeared in section 3.5 before [46-57] in section 4.

·        The paper limited the VLC applications into two areas, positioning and vehicular communications. Other applications of VLC are required to be included.

Reviewer 2 Report

In this work, the Advances in Visible Light Communication Technologies and

Applications are reviewed. To enhance the paper, I have some considerations as follows:

1-    In section 3.3, it would be better if the authors have more extensive discussions on physical-layer security, at least making the cryptanalysis more formal. Physical-layer security has emerged as a promising complement to conventional encryption techniques to counter eavesdropping. The two main categories of physical layer security techniques are, (1) keyless security techniques, and (2) key-based security techniques. In this work, the authors consider the first category that provides secrecy for the wireless system by exploiting the properties of the wireless channel. In this technique, the secret key is not required. What about the second category? The second technique “key-based security techniques require a secret key to provide secrecy. Although these techniques provide less security than the first, They are less complicated than keyless security approaches. The authors should consider the recent work in this field such as:

(1)  Y. M. Al-Moliki, M. T. Alresheedi, Y. Al-Harthi and A. H. Alqahtani, "Robust Lightweight-Channel-Independent OFDM-Based Encryption Method for VLC-IoT Networks," in IEEE Internet of Things Journal, vol. 9, no. 6, pp. 4661-4676, 15 March 15, 2022.

(2)  R. Melki, H. N. Noura and A. Chehab, "Efficient & Secure Physical Layer Cipher Scheme for VLC Systems," 2019 IEEE 90th Vehicular Technology Conference (VTC2019-Fall), 2019, pp. 1-6.

(3) Y. Al-Moliki, M. Alresheedi and Y. Al-Harthi, "Design of physical layer key generation encryption method using ACO-OFDM in VLC networks", IEICE Trans. Commun., vol. E103-B, no. 9, pp. 969-978, 2020.

(4) Y. M. Al-Moliki, M. T. Alresheedi and Y. Al-Harthi, "Chaos-based physical-layer encryption for OFDM-based VLC schemes with robustness against Known/chosen plaintext attacks", IET Optoelectron., vol. 13, no. 3, pp. 124-133, 2019.

(5) Z. Wang, Z. Wang and S. Chen, "Encrypted image transmission in OFDM-based VLC systems using symbol scrambling and chaotic DFT precoding", Opt. Commun., vol. 431, pp. 229-237, Jan. 2019.

(6) Y. Al-Moliki, M. Alresheedi and Y. Al-Harthi, "Improving availability and confidentiality via hyperchaotic baseband frequency hopping based on optical OFDM in VLC networks", IEEE Access, vol. 8, pp. 125013-125028, 2020.

2-    Also, the authors should consider the recent modulation techniques proposed for VLC such as PWM. The authors may consider the following reference:

(1) M. A. S. Sejan, R. P. Naik, B. -G. Lee and W. -Y. Chung, "A Bandwidth Efficient Hybrid Multilevel Pulse Width Modulation for Visible Light Communication System: Experimental and Theoretical Evaluation," in IEEE Open Journal of the Communications Society, 2022.

(2) X. -Y. Xu, Q. Zhang and D. -W. Yue, "Orthogonal Frequency Division Multiplexing With Index Modulation Based on Discrete Hartley Transform in Visible Light Communications," in IEEE Photonics Journal, vol. 14, no. 3, pp. 1-10, June 2022, Art no. 7330310, doi: 10.1109/JPHOT.2022.3174283.

(3) O. Saied, X. Li and K. M. Rabie, "DFT Spread-Optical Pulse Amplitude Modulation for Visible Light Communication Systems," in IEEE Access, vol. 10, pp. 15956-15967, 2022.

(4) Other related recent references.

Reviewer 3 Report

The authors have comprehensively investigated the VLC technologies and applications, which is inspiring. There are a few questions and comments that should be necessarily replied to before the publication.

1. It would be more appealing include how to avoid the disturbance of the signal since visible lights possess lower wavelengths, which could be easier scattered by small particles. 

2. Could some examples be listed for scenarios that are not suitable for using RF but applicable for using VLC?

3. It might be interesting to show the communication distance range for visible light communication.

4. Please adjust the text boxed in Fig 6 as some of them were partially displayed.

5. For content in tables, it would be visually pleasing for the audience to see a different formate of the title row(such as highlighting or shading the first row) unless the format in the manuscript is required by the formate of the journal.

Round 2

Reviewer 2 Report

This version of the paper can be published now.